rsob.royalsocietypublishing.org

Subject Area:
developmental biology

Keywords:
Gata3, haematopoiesis, development

Author for correspondence:
Katrin Ottersbach
e-mail: katrin.ottersbach@ed.ac.uk

# The multi-faceted role of Gata3 in developmental haematopoiesis

Nada Zaidan[1,2] and Katrin Ottersbach[1]

[1]MRC Centre for Regenerative Medicine, University of Edinburgh, Edinburgh EH16 4UU, UK
[2]King Abdullah International Medical Research Centre, Ministry of National Guard Health Affairs, Riyadh, Kingdom of Saudi Arabia

 KO, 0000-0002-6880-4895

The transcription factor Gata3 is crucial for the development of several tissues and cell lineages both during development as well as postnatally. This importance is apparent from the early embryonic lethality following germline Gata3 deletion, with embryos displaying a number of phenotypes, and from the fact that Gata3 has been implicated in several cancer types. It often acts at the level of stem and progenitor cells in which it controls the expression of key lineage-determining factors as well as cell cycle genes, thus being one of the main drivers of cell fate choice and tissue morphogenesis. Gata3 is involved at various stages of haematopoiesis both in the adult as well as during development. This review summarizes the various contributions of Gata3 to haematopoiesis with a particular focus on the emergence of the first haematopoietic stem cells in the embryo—a process that appears to be influenced by Gata3 at various levels, thus highlighting the complex nature of Gata3 action.

## 1. Introduction

Haematopoiesis refers to the formation of blood cellular components. All these components originate from multipotent haematopoietic stem cells (HSCs), which form the foundation of this process. The system is traditionally seen as hierarchical, with the multipotent HSC being the mother cell that gives rise to and differentiates into multipotent and unipotent intermediate progenitor cells, resulting in the production of functional mature blood cells, although in recent years exceptions have been discovered that have an HSC-independent origin (reviewed in [1]). As such, HSCs have unique characteristics that in combination distinguish them from other more mature cells: (i) self-renewal ability; (ii) high proliferation ability; (iii) long-term activity; and (iv) potential to differentiate into all the different haematopoietic lineages. All these characteristics make those cells the most clinically relevant cells for transplants.

Haematopoiesis is a complex and intricate process that is governed by a large number of signalling pathways and transcription factors. The transcription factors themselves are often organized in multi-gene families and play essential roles in activating target genes of specific cell fates and in repressing target genes of alternative cell fates. The GATA family of transcription factors are such master regulators. This family has six members in vertebrates, and the disruption of each of the *Gata* genes, with exception of *Gata5*, causes embryonic lethality in mice. They are grouped into haematopoietic (Gata1, Gata2, Gata3), and endodermal (Gata4, Gata5, Gata6) subgroups (recently reviewed in [2]). Each GATA transcription factor is highly conserved across vertebrates. For example, Gata3 homologues are found in human, mouse, rat, chimpanzee, dog, chicken, frog and zebrafish. Gata3 shares 97% of its amino acid identity between mouse and human. Within the GATA family, members share varying degrees of homology. For example, while GATA2 and GATA3 are about 55% homologous at the amino acid level, GATA3 and GATA4 are only 20% homologous.

The largest degree of conservation is found in the zinc finger domains, which are about 80% homologous among all six members. The two zinc fingers bind to different sequences and each has a unique function. The C-finger binds to the GATA consensus site, WGATAR, although in a genome-wide ChIP-seq experiment, Fujiwara *et al.* [3] reported that GATA proteins preferred binding to WGATAA sequences *in vivo*. However, the abundance of these motifs is such that the binding of GATA factors to specific DNA loci cannot be inferred from the presence of the sequence alone. Instead, it has become clear that occupancy of specific DNA sites by GATA factors is dependent on the presence of other co-regulatory proteins which can facilitate recruitment of GATA factors to specific protein complexes or regulate GATA activity through post-translational modifications (reviewed in [4]).

The N-finger facilitates the interaction with such co-regulatory proteins, but can also bind specific, yet distinct, DNA sequences [3,5–7]. This combination of DNA sequence-specific binding and recruitment to chromatin via a dynamic range of protein complexes resulting in both transcriptional activation and repression allows GATA factors to participate in a large repertoire of different processes that are highly cell context-dependent. In the case of Gata3, this has been extensively studied in lymphoid development, regulated by Gata3 at various stages, which has recently been comprehensively reviewed [8]. In addition, in a crystallographic structural study by Bates *et al.* [9] for Gata3, it was observed that GATA family members could bind to DNA by either homo- or hetero-dimerizing, by forming a dimer with other GATA factors through their two C-fingers.

All GATA factors have both distinct and common biological roles and biochemical characteristics, and all have a restricted expression pattern, which is controlled by tissue-specific enhancer elements. The regulatory elements that control Gata3 expression have been well characterized through the identification of enhancers that drive its expression in the urogenital system, the central nervous system, the endocardium and in natural killer (NK)/T-cells, with the NK/T-cell-specific element being located 280 kb downstream of the *Gata3* gene [10,11]. However, in some cases, the functions of GATA factors are interchangeable [12]. For example, Gata1, Gata2, Gata3 and Gata4 can activate interleukin-4 (Il4) and Il5 expression in T-cells, which are classically target genes for Gata3, and repress the activation of interferon-γ [13]. Moreover, a *Gata3* knock-in can partially rescue erythrocyte defects in *Gata1* null mice; however, Gata3 cannot fully rescue the phenotype of *Gata1* null mice, indicating that each GATA factor maintains its unique functions [14,15].

# 2. The three haematopoietic GATAs

While Gata4, Gata5 and Gata6 drive differentiation of mesoderm- and endoderm-derived tissues and are therefore critical for the development of heart and lung, the first three members of the GATA family are involved in the differentiation of mesoderm- and ectoderm-derived tissues and play essential roles in the development and maintenance of the haematopoietic system. Very broadly speaking, the main function of Gata1 is cell fate determination at an early branch point in the haematopoietic tree, where it induces megakaryocyte and erythrocyte development, while preventing granulocyte-monocyte and lymphoid commitment. However, it is also

expressed further downstream in common lymphoid and myeloid progenitors, mast cells and eosinophils [16,17]. The most critical role of Gata2 is the formation and maintenance of HSCs [18,19], although it has additional functions in specific blood lineages as discussed below. Gata3 is crucial for the development of several lymphoid lineages (reviewed in [8]) and early definite haematopoietic stem and progenitor cells [20,21], which will be discussed further below.

The haematopoietic group within the GATA factors control each other's expression during development in different cells, and are capable of functioning consecutively during cell specification and lineage commitment in a process called a GATA switch. GATA switch refers to instances where one GATA factor is replaced by another GATA at the chromatin site. GATA switches occur at many functionally critical loci during development, including those that control the expression of regulators of haematopoiesis, such as Gata2 itself [22]. Gata2 is a direct target of Gata1; however, in the absence of Gata1, it can bind to a target region upstream of its own promoter, which activates transcription and induces histone acetylation. However, when Gata1 is expressed, Gata2 is displaced by Gata1 at its chromatin site, which activates erythropoiesis [23,24] (and reviewed in [4,22]).

## 2.1. Gata1

The essential role of Gata1 in erythropoiesis was demonstrated in Gata1-deficient mice which suffer from early embryonic death (E10.5–11.5) and an inablility to complete primitive and definitive erythroid differentiation [25,26]. Gata1 is expressed in HSCs and common myeloid and/or lymphoid progenitors. It is also crucial for the development of the megakaryocyte lineage [27] and for the survival of erythrocyte precursors [28,29]. Gata1 downregulates cofactors that are necessary for granulocyte–monocyte and lymphoid development, including Spi1 (PU.1), Il7 and Pax5 [30,31], while promoting megakaryocyte and erythrocyte commitment. Gata1 is also expressed in eosinophils and mast cells, where it plays a role in their terminal differentiation [16,17]. Functionally, Gata1 is involved in cell cycle regulation. In the context of erythroid maturation, it was shown to induce G1 arrest by targeting a number of key cell cycle regulators, which allows the cells to undergo maturation, driven by a Gata1-dependent erythroid gene expression programme [32].

## 2.2. Gata2

Gata2 is a master regulator of haematopoiesis. It is expressed in HSCs, multipotent haematopoietic progenitors, megakaryocytes, erythroid precursors, eosinophils and mast cells. Its deletion leads to embryonic death at E10.5 and a complete disruption of definitive haematopoiesis [33]. This is at the level of HSCs, as Gata2 is required for their emergence (as discussed further below) and their subsequent survival in a dose-dependent fashion [18,19,34]. However, while Gata2 is required for the proliferation and survival of multipotent haematopoietic progenitors and mast cell formation, it is dispensable for the terminal differentiation of erythroid cells and macrophages [35].

## 2.3. Gata3

Gata3 has been extensively studied in the context of innate and adaptive lymphoid development, where it regulates

rsob.royalsocietypublishing.org   Open Biol. 8: 180152

rsob.royalsocietypublishing.org Open Biol. **8**: 180152

differentiation and cell fate determination at various levels (for an extensive recent review see [8]). Specifically, it was found to be essential for the development, maintenance, survival and proliferation of early T-cell progenitors, as ES cells with a deleted *Gata3* gene were able to contribute to the B-cell, myelomonocytic and erythroid lineages but not thymocytic or T-cell lineages in chimera studies [36]. It was subsequently demonstrated that Gata3 is required to induce and seal the T-cell fate in early lymphoid progenitors, while repressing the B lymphoid programme [37,38]. Within the T-cell lineage, Gata3 is a master regulator of T helper type 2 cells (Th2). It regulates the differentiation of Th2 cells by controlling genes that encode Th2 cytokines Il4, Il5 and Il13 [39]. It appears, however, that Gata3 levels need to be carefully controlled throughout thymocyte development as levels that are too high are cytotoxic, and levels that are too low cause developmental failure [40]. Indeed, enforced expression of Gata3 in the T-cell lineage caused a maturation arrest in the cytotoxic T-cell lineage and promoted the formation of thymic lymphoblastoid tumours [41], and overexpression of Gata3 at early fetal thymocyte stages redirected their differentiation towards the mast cell lineage [42].

Within the branch of innate immunity, Gata3 is central to the development of the recently discovered innate lymphoid cells (ILCs), especially the ILC2 lineage [43,44], a subset of ILC3 cells [45] and a subset of ILC1-derived, tissue-resident NK cells [46,47]. Thus, Gata3 is not necessary for the development of classic (NK) cells, but is crucial for a specialized subset of them. It is important for the terminal differentiation of NK cells and their exit from the bone marrow, and is crucial for the maintenance of liver-resident NK cells [48].

The importance of GATA3 in lymphoid development and function is further highlighted by the fact that GATA3 has also been implicated in T-cell acute lymphoblastic leukaemia (T-ALL). Together, T-cell acute lymphocytic leukaemia 1 (*TAL1*), *RUNX1* and *GATA3* form a positive interconnected auto-regulatory loop that directly activates the *MYB* oncogene, thus reinforcing and stabilizing the oncogenic programme that contributes to malignant transformation [49]. In addition, whole genome sequencing of patients with early T-cell precursor acute lymphoblastic leukaemia (ETP-ALL), an aggressive subtype of T-ALL, has revealed GATA3 inactivating lesions disrupting haematopoietic development [50]. GATA3 has also been linked to other types of lymphoid malignancies. In a genomic profiling study, a GATA3 single-nucleotide polymorphism genotype has been identified in a subtype of childhood acute lymphoblastic leukaemia (ALL), Philadelphia chromosome-positive ALL (Ph-like ALL), that has been associated with early treatment response, higher risk of relapse and overall poor prognosis [51]. And in anaplastic large cell lymphoma, the absence of the GATA3 protein in addition to the presence of suppressive histone (H3K27) trimethylation at the *GATA3* promoter suggests epigenetic regulation of GATA3 as a mechanism involved in disease pathogenesis [52].

Gata3 is also highly expressed in the long-term repopulating HSC (LT-HSC) population [53–55]. Using *Gata3*-null mice (deleted postnatally with Mx1-Cre), Ku *et al*. [56] have shown that *Gata3* deletion results in the production of lower numbers of adult LT-HSCs, and that a lower number of these Gata3-deficient LT-HSCs are in cycle. This suggests that Gata3 is necessary for maintaining normal numbers of LT-HSCs, and that it regulates their entry into the cell cycle. However, by using a conditional *Gata3* knockout mouse line crossed to a

**Table 1.** Tissue-specific functions of Gata3.

| system | function | reference |
|---|---|---|
| skin and hair | generation of skin selective barrier | [62–64] |
| | promotion of progenitor differentiation in the hair follicle and proper hair structure | |
| kidney | nephric duct development | [65,66] |
| fat | inhibition of adipocyte differentiation | [67] |
| inner ear | cochlea morphology | [68] |
| mammary gland | mammary gland morphology | [69,70] |
| haematopoietic system | T-cell development | [36] |
| | Th2 commitment | [39] |
| | ILCs | [43–45] |
| | NK subsets | [46–48] |
| | Cell cycle regulation in adult HSCs | [56,58] |
| | non-cell autonomous role in embryonic HSC production | [20] |
| SNS | essential for the production of catecholamines | [71,72] |
| | promotes survival of sympathetic neurons in both adults and embryos | |

Vav-Cre line, Buza-Vidas *et al*. [57] have shown that deletion of *Gata3* from HSCs after their emergence in the embryo does not affect the ability of HSCs to expand normally and that their numbers remain unaffected in the bone marrow after birth. Moreover, they reported that *Gata3* deletion does not affect the ability of HSCs to self-renew [57].

More recently, Frelin *et al*. [58] published data suggesting that Gata3 controls the balance of LT-HSC self-renewal and differentiation by regulating their reprogramming from LT-HSC to intermediate term HSCs (IT-HSC). IT-HSCs are important for maintaining blood counts at steady state. They differ from LT-HSCs in that they are able to generate myeloid and erythroid progeny for 12 weeks [53,54,59], are more abundant than LT-HSCs (three times higher) [53,60], are more proliferative, and exit the quiescent state of the cell cycle more frequently than LT-HSCs: every 10–20 days compared to 50–100 days in LT-HSCs [61]. However, the precise function of Gata3 in adult HSCs requires further investigation.

# 3. *Gata3* in development

Gata3 plays a major role in cell lineage specification and development in a variety of cells, tissues and organs during embryogenesis (table 1), including adipocytes [67], kidney [65], mammary gland [69], skin [62,73] and sympathetic nervous system (SNS) [63,71,74]. Not only does the study of

Gata3 activity in these different systems provide us with important clues about the molecular details of Gata3 function, it has also demonstrated that due to the close proximity of developing systems in the embryo, Gata3 action in one tissue can influence the development of a neighbouring tissue [20].

## 3.1. Skin and hair

Gata3 is essential for stem cell lineage determination in the skin. It is expressed at the onset of inner root sheath (IRS) cell specification in hair follicles. When *Gata3* was deleted using a LacZ knock-in, IRS progenitors failed to differentiate to form IRS, leading to the production of a defective hair structure [73]. In addition, Gata3 is the most highly expressed member of the GATA family in the interfollicular epidermis. The specific deletion of *Gata3* from the epidermal layer, using a *keratin-14-Cre* (*K14-Cre*) mouse line, proved to be prenatally lethal due to impairment of the skin selective barrier. Those mice showed defects in skin differentiation, abnormal hair follicle organization and delayed hair growth and maintenance. Genomic analysis of the mice revealed defective lipid biosynthesis. This could be attributed to the loss of lipid acetyltransferase gene (*Agpat5*), a gene that is a direct target of Gata3 [62,64]. Interestingly, Gata3 has previously been linked to adipogenesis where it was shown to inhibit adipocyte differentiation [67].

## 3.2. Kidney

Gata3 is the only GATA factor that is expressed in the urogenital system prior to E12.5. It is necessary for the normal development of the nephric duct. Using a *HoxB7-Cre* transgenic line, Grote *et al.* [65,66] specifically deleted Gata3 from the nephric duct, which resulted in severe abnormalities in the urogenital system and revealed that Gata3 is required to prevent ectopic metanephric kidney duct formation and premature cell differentiation. Additionally, it was reported that Gata3 haploinsufficiency resulted in renal dysplasia.

Gata3 has also been implicated in clear cell renal cell carcinoma (cc-RCC), the most common subtype of RCC. Cooper *et al.* [75] uncovered that when Gata3 expression was downregulated by promoter hypermethylation, it resulted in decreased expression of type III TGF-β receptor (TβRIII), which is a betaglycan protein with tumour suppressor features [75].

## 3.3. Inner ear

Gata3 is widely expressed in several cell types during ear development, including inner hair cells, outer hair cells as well as supportive cells [76–78]. As a consequence, the entire cochlea of the inner ear shows significant degeneration in mice heterozygous for *Gata3*, which leads to hearing loss [68]. This is mirrored in patients with HDR syndrome, who have only one functional copy of *GATA3* and who suffer from hypoparathyroidism, deafness and renal defects [79,80].

## 3.4. Mammary gland

Gata3 is crucial for mammary gland development. It is the transcription factor with the highest expression in the mammary epithelium as shown by genome-wide transcript analysis [70]. When *Gata3* was specifically deleted from the mammary epithelium at the onset of puberty, using the murine mammary

tumour virus (MMTV) promoter-Cre recombinase (MMTV-Cre), the mammary glands failed to develop terminal end buds (TEBs), resulting in abnormal ductal structures [69,70].

Its crucial role in the mammary gland is supported by the detection of GATA3 mutations in around 10% of human breast cancers. While the range of somatic mutations is varied, they cluster mainly in the highly conserved C-terminal second zinc finger [81]. Data from *in vitro* and *in vivo* studies suggest that GATA3 acts as a tumour suppressor gene. In a murine luminal breast cancer model, the loss of *Gata3* resulted in tumour progression and tumour dissemination [82]. More specifically, GATA3 expression was shown to inhibit breast cancer growth and pulmonary metastasis by repressing metastasis-associated genes such as *ID1, ID3, KRTHB1, LY6E* and *RARRES3* [83], and restoration of Gata3 expression in a breast cancer mouse model induced breast cancer differentiation and suppressed its dissemination [82]. Moreover, GATA3 was found to promote the expression of microRNA-29b (miR-29b), which in turn induces differentiation, suppresses metastasis and changes the tumour microenvironment [84]. In addition, low expression of GATA3 was associated with a poor survival rate and more aggressive disease, whereas GATA3-expressing breast cancer patients had a better prognosis, were less likely to relapse, and had a better overall survival rate [85]. The involvement of GATA3 in breast cancer, however, is complex as it was also shown to promote the growth of oestrogen-responsive tumours through direct binding to and activation of the oestrogen receptor α (ERα) gene [86]. ERα-positive tumours display a more differentiated phenotype and are generally less aggressive, which may be another reason why GATA3 expression in breast cancer is associated with a more favourable prognosis [87–89].

## 3.5. Sympathetic nervous system

Gata3 is essential for the development of the SNS [63,71,72,74,90]. In fact, this is the reason why *Gata3* deletion is embryonically lethal at around E11.5. This lethality was attributed to noradrenaline deficiency in the SNS and could be pharmacologically rescued by feeding the mothers DOPS, a synthetic catecholamine intermediate [63]. It was subsequently confirmed that Gata3 is essential for the production of catecholamines, the SNS mediators, through controlling the expression of tyrosine hydroxylase, the enzyme required for catecholamine synthesis. It also plays a major role in the survival of sympathetic neurons in both adults and embryos [71,72].

# 4. *Gata3* in haematopoietic stem cell emergence

The process of HSC emergence is highly conserved in vertebrates and is closely linked to vascular development (reviewed in [91,92]). It has been most intensely studied in the intra-embryonic aorta–gonad–mesonephros (AGM) region, where the first directly transplantable HSCs emerge at E10.5 in the mouse [93,94]. Haematopoietic stem and progenitor cell emergence involves activation of a haematopoietic transcriptional programme in a subset of endothelial cells, termed haemogenic endothelial cells, within the major arterial vessels of the embryo, such as the dorsal aorta within the AGM [95–97]. These haemogenic endothelial cells then undergo major structural and morphological changes that allow them

rsob.royalsocietypublishing.org Open Biol. 8: 180152

rsob.royalsocietypublishing.org    Open Biol. 8: 180152

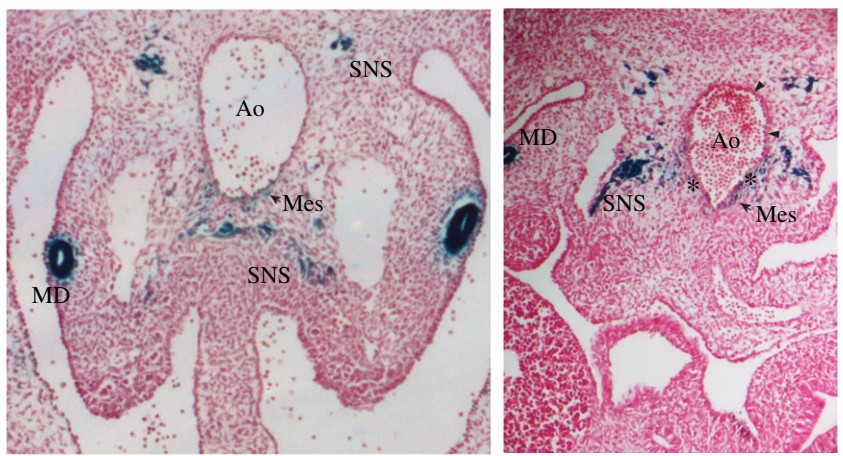

**Figure 1.** Gata3 expression in the AGM region. Images show cryosections of Gata3$^{+/lz}$ [114] E11.5 embryos stained with X-Gal for β-galactosidase activity (blue) and counterstained with Neutral Red. Ao, aorta; MD, mesonephric duct; Mes, mesenchyme; SNS, sympathetic nervous system; asterisks indicate Gata3 staining underneath haematopoietic clusters; arrowheads point to individual Gata3-expressing endothelial cells.

to round up and detach from the endothelium as haematopoietic stem and progenitor cells. Localized production of de novo blood cells can be detected in tissue sections as clusters of cells co-expressing endothelial and haematopoietic markers that are attached to the endothelium, and has recently been captured by live imaging [98–100]. This endothelial-to-haematopoietic transition (EHT) is acutely dependent on the transcription factor Runx1 [101]. In its absence, haemogenic endothelial cells undergo apoptosis [100], no intra-aortic clusters are formed [102–104] and the generation of definitive haematopoietic stem and progenitor cells is disrupted [105,106].

The recent development of an elegant *ex vivo* co-aggregation explant culture system has revealed additional maturation stages that haemogenic endothelial cells have to undergo before they become fully functional HSCs [107–109]. There are at least three intermediate states as haemogenic endothelial cells mature first into pro-HSCs, then type I pre-HSCs and eventually via type II pre-HSCs into adult-type HSCs that can directly repopulate adult recipients. These stages can be distinguished temporally and through the sequential upregulation of the haematopoiesis-associated cell surface markers CD41, CD43 and CD45 [107].

A day after their emergence in the dorsal aorta and associated vitelline and umbilical arteries, HSCs are also detected in the yolk sac and the placenta [93,110–112]. Their presence in these tissues is only temporary as they eventually go on to colonize the fetal liver which becomes the major haematopoietic organ in the embryo from E12.5 until birth when HSCs relocate to the bone marrow.

Gata3 is expressed in the sites of intraembryonic haematopoietic cell generation in the mouse (AGM and its precursor, the para-aortic splanchnopleura) [20,113] (figure 1), zebrafish [115], chicken [116], human [117] and in *Xenopus* [118,119]. In the mouse embryo, Gata3 is expressed at low level at E8.5 in the splanchnic mesoderm [113]. However, by E11.5 Gata3 is expressed at high levels throughout the embryo. At this stage, *Gata3* deletion was embryonically lethal, death occurring concomitantly with the onset of definitive haematopoiesis in the fetal liver [21]. *Gata3* knockout embryos were shown to have growth retardation, along with severe deformities in spinal cord and brain, massive internal haemorrhage, anaemia and defective liver haematopoiesis, i.e. definitive haematopoiesis, suggesting that Gata3 is essential for the development of various systems [21]. Yolk sac (YS) haematopoiesis was

normal, which corresponds with the fact the Gata3 is not expressed in the YS [113]. Specifically, in an *in vitro* culture system, the colonies that resulted from the YS of Gata3 knockout embryos, compared to their wild-type and heterozygous littermates, were normal, indicating the maintenance of primitive erythropoiesis. However, the colony numbers from the fetal liver of the knockout embryos were low compared to the wild-type and heterozygous littermates, indicating that Gata3 disruption severely affects definitive haematopoiesis [21].

## 4.1. The sympathetic nervous system as part of the haematopoietic stem cell niche

The observation that definitive haematopoiesis is affected in Gata3$^{-/-}$ embryos [21] and that Gata3 is expressed in the AGM [113] (figure 1) suggested that Gata3 may also be involved in HSC emergence. Furthermore, expression analyses and sorting of specific cell populations followed by transplantation into immunocompromised mice led to the proposal that defined structures located ventrally to the dorsal aorta, termed sub-aortic patches, contain HSCs and/or their precursors and that these expressed Gata3 [113,120]. The cells in these patches were also found to co-express Gata2.

Gata3 was also identified as a potential AGM haematopoiesis regulator in expression profiling studies [121]. This study comprised three comparisons: (i) HSC containing region (middle part of the dorsal aorta) versus a region without HSCs (caudal and rostral part of the aorta); (ii) the microenvironment of HSCs before and after their emergence, i.e. the aorta with the immediate mesenchyme of E9–E10 versus E11; and (iii) using *Ly-6A GFP* transgenic embryos, which express GFP in all embryonic HSCs and their precursors, populations enriched for HSCs (E11 Ly-6A GFP+ cells) or their precursors (E9 Ly-6A GFP+ cells) were compared. *Gata3* was found to be upregulated in two of the three comparisons (ii and iii), i.e. in tissues surrounding the dorsal aorta specifically at the time of HSC emergence in the AGM and in HSC-enriched populations: E11 Ly-6A GFP+ cells.

A role for Gata3 in HSC production in the AGM was subsequently confirmed [20]. Germline-deleted Gata3$^{-/-}$ AGMs contained fewer Ly-6A-GFP+ aortic endothelial cells and showed reduced intra-aortic cluster formation. Most importantly, HSC activity in transplantation assays was severely

rsob.royalsocietypublishing.org   Open Biol. **8**: 180152

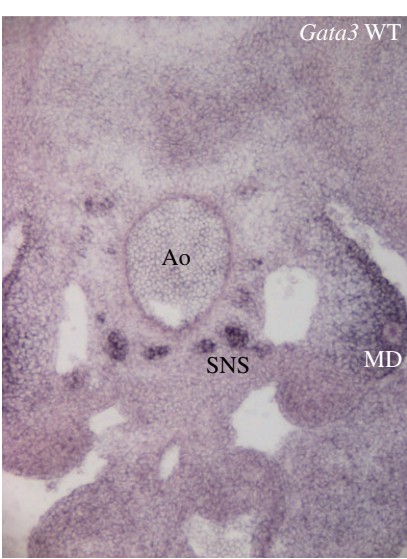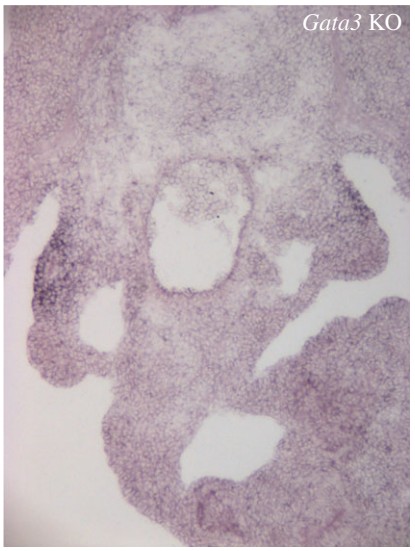

**Figure 2.** Gata2 expression in *Gata3* wild-type (WT) and knockout (KO) AGMs. Images of *in situ* hybridization with a *Gata2* riboprobe on cryosections from E11.5 *Gata3*$^{+/+}$ (left) and *Gata3*$^{-/-}$ [21] (right) embryos. Ao, aorta; MD, mesonephric duct; SNS, sympathetic nervous system.

reduced in *Gata3*$^{-/-}$ and *Gata3*$^{+/-}$ AGMs. Interestingly, transplantation of Gata3-LacZ+ and Gata3-Lacz− AGM cells clearly assigned repopulation activity to the Gata3-LacZ− population, strongly suggesting that Gata3 is not expressed in newly formed HSCs, but performs a non-cell autonomous role via the AGM haematopoietic microenvironment [20].

One of the components of the AGM haematopoietic microenvironment turned out to be the co-developing SNS. It had previously been reported that the SNS plays a major role in the mobilization [122,123] and proliferation [124] of adult HSCs. The fact that Gata3 deletion affected both the SNS [63] as well as HSC production [20] in the AGM suggested the intriguing possibility that a functional interplay between the haematopoietic system and SNS already occurred at the time when these first develop during embryogenesis. Indeed, it was then demonstrated that external provision of catecholamines to Gata3-deficient embryos rescued the HSC defect, confirming that Gata3 regulates HSC numbers through catecholamine production [20]. This indicates that HSC emergence in the AGM should be investigated as a part of a whole developmental process that is influenced by neighbouring tissues [20,121]. It also suggests that the previously described sub-aortic patches [113,120] may, in fact, have been cells of the SNS as they are known to co-express Gata3 and Gata2 [20,71] (figures 1 and 2). This does not rule out, however, that there may be individual Gata2 and Gata3 expressing mesenchymal HSC precursors [125].

## 4.2. A role for Gata3 in haematopoietic stem cell precursors

In addition to its role in the AGM HSC microenvironment (via the SNS) described above, there is also evidence that Gata3 may contribute to the specification of the definitive haematopoietic lineage. Manaia *et al.* [113] analysed the expression of Gata3 and Lmo2 in mouse embryonic development to understand the mechanisms involved in the generation of definitive HSCs. Interestingly, they found that Gata3 and Lmo2 are expressed concomitantly in the caudal embryonic mesoderm where haematopoietic cluster-bearing vessels develop, suggesting an involvement in cell fate determination. Another observation of their study was that Gata3

expression is restricted to sites involved in definitive haematopoiesis. Gata3 is expressed in the environment from which intraembryonic precursors emerge, and in the developing haematopoietic sites before their colonization. Gata3 was expressed both in the thymic rudiment until the first migrants arrived, and in the septum transversum before it gives rise to the fetal liver. However, no CD45+ haematopoietic cells were present within those sites at that developmental stage [113].

More recently, *Gata3* was also found to be upregulated in a subset of E10.5 endothelial cells that express GFP under the control of the Runx1 + 23 enhancer element (23GFP+) that was described to mark haemogenic endothelial cells [97], indicating a possible involvement in HSC and progenitor generation. In addition, Gata3 expression increased upon Notch1 signalling induction, which expands the haemogenic endothelial population, and enhanced their haematopoietic potential [126]. Indeed, we found Gata3 expression in individual endothelial cells, often in the vicinity of haematopoietic clusters (figure 1); however, further studies are required to determine if Gata3 plays a functional role in haemogenic endothelium and the EHT.

## 4.3. Gata3 expression in other compartments of the embryonic haematopoietic stem cell microenvironment

Gata3 expression is also found in individual cells in the subaortic mesenchyme of the AGM region (figure 1). Expression in this stromal compartment is restricted to the ventral side of the dorsal aorta, where HSCs are preferentially located [127]. In fact, Gata3-expressing cells were often detected just underneath intra-aortic haematopoietic cell clusters [20]. This mesenchymal expression disappears after E12.5, when intraembryonic clusters cease to be generated [113]. A number of known regulators of HSC emergence are expressed in cells of the sub-aortic ventral mesenchyme including Bmp4, Scf, Runx1, Bmper, Thpo and Dlk1, demonstrating that this cell compartment forms an important embryonic HSC niche [128–133]. Furthermore, cells with mesenchymal stem/stromal cell potential have been detected specifically in the AGM at the time of HSC emergence

rsob.royalsocietypublishing.org  Open Biol. **8**: 180152

[134]. The sub-aortic mesenchymal cell compartment, however, is a very heterogeneous population, and it is currently unknown which cells participate in the niche for emerging HSCs and whether this compartment contains and may even be maintained by mesenchymal stem cells. It is also currently not clear whether Gata3 plays an important role in the mesenchymal stroma of the AGM.

Gata3 is expressed in one further cell compartment in the AGM region, the mesonephric ducts within the urogenital ridges (UGRs) [20] (figure 1). It was recently demonstrated that UGRs do not contain HSCs or their precursors, but their presence promotes HSC formation in the neighbouring aorta in co-aggregation studies, indicating that UGRs also form part of the HSC supportive microenvironment [133]. Yet, while it has been established that Gata3 expression in the UGRs is required for kidney development [65], it is currently unknown whether it promotes the production of HSC-supportive factors in these structures. However, the fact that Gata3 is expressed in several different cell types in the AGM relevant to haematopoiesis highlights its complex involvement in the production of the first HSCs.

# 5. Concluding remarks

Gata3 is essential for the development of several types of cells, organs and tissues (table 1), and its disruption during development results in severe defects and impairment in those systems, leading to embryonic death at midgestation following germline deletion [21]. Analysis of the phenotypes of *Gata3* deletion in these different systems has revealed that Gata3 is often expressed in the stem and progenitor compartment where it regulates cell fate determination and differentiation. In the developing kidney [65,66] and preadipocytes [67], it seems to prevent premature differentiation, whereas in skin [73] and mammary gland [69] it promotes progenitor differentiation. Overall, however, its function is to ensure correct tissue morphogenesis. Considering this crucial function, it is therefore not surprising that Gata3 has been implicated in a number of different cancer types.

How Gata3 functions at the molecular level is not well understood in many of these systems. One common underlying theme may be regulation of the cell cycle as has been suggested for the role of Gata3 in adult HSCs [56,58]. Gata3 may allow tissue-specific progenitors to differentiate by blocking their cell cycle, and this may also be how haemogenic endothelial cells can then undergo the morphological changes as they become blood cells. However, it is also clear that Gata3 activates tissue-specific genes such as tyrosine hydroxylase in the SNS and lipid acetyltransferase in the skin. Some of the genes that Gata3 activates during metastasis may also be relevant for the EHT.

GATA factors have often been observed in common complexes and have been seen to cross-regulate each other, with the GATA switch being such an example. Within the AGM, Gata2 and Gata3 expression overlaps in several cell compartments (figures 1 and 2) and they are both involved in HSC production in the AGM, but the nature of their interaction appears to be tissue-specific. Both are expressed in the urogenital ridges, but while Gata3 is expressed specifically in the mesonephric ducts, Gata2 expression is found in the tissue directly surrounding the ducts. Interestingly, however, the expression of Gata2 around the ducts disappears in $Gata3^{-/-}$

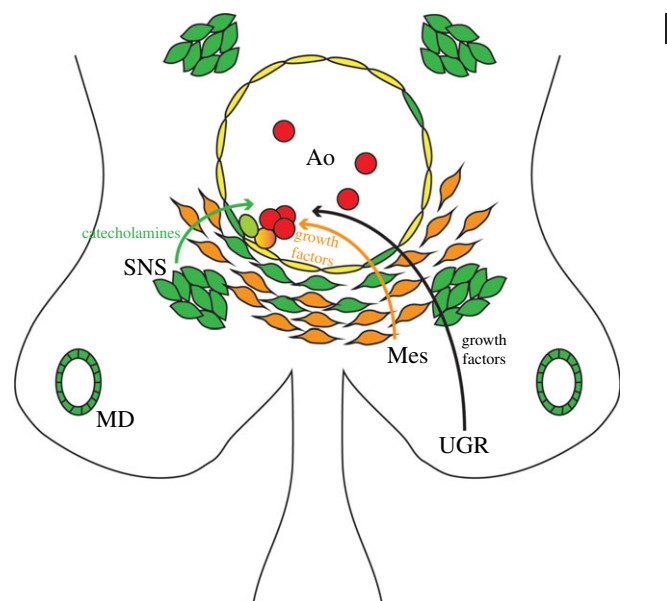

**Figure 3.** Gata3 involvement in AGM haematopoiesis. Schematic diagram of a transverse section through an E11.5 AGM region, highlighting the cell compartments that express Gata3. Gata3-positive cells (green) are found within the endothelial layer (yellow) of the dorsal aorta (Ao), in the mesonephric duct (MD), within the subaortic mesenchyme (orange; Mes) and in the sympathetic nervous system (SNS). Blood cells are shown in red. The light green cell depicts the putative involvement of Gata3 in the endothelial-to-haematopoietic transition. Curly arrows illustrate contributions made by the different components of the microenvironment to EHT/HSC support, of which only catecholamines are currently known to be Gata3-dependent. UGR, urogenital ridges.

embryos [20] (figure 2). Both have been detected in the haemogenic endothelium, but the deletion of Gata2 affects the development of definitive haematopoietic stem and progenitor cells much more severely, suggesting that it may act upstream of Gata3 here or performs a much more crucial function in the EHT. In the SNS, on the other hand, Gata3 is clearly upstream of Gata2, as Gata2 expression in sympathoadrenal cells disappears in *Gata3*-null embryos, while its expression in the aortic endothelium remains [20] (figure 2).

Very little is currently known about the upstream regulators of Gata3 in the different AGM cell compartments. Phox2b is required for Gata3 expression in the SNS [71]. Considering the crucial roles of Gata2 and Runx1 in the EHT, these two transcription factors may well be upstream of Gata3 in haemogenic endothelial cells. In addition, Notch1 was shown to induce Gata3 in an embryonic stem cell model of the EHT [126]. As Notch1 is also crucial for HSC emergence in the AGM [135], it is likely that Gata3 dependence on Notch signalling is conserved *in vivo*.

Current data have shown that Gata3 plays several roles in the embryonic and adult haematopoietic system. However, the similarities and differences in these roles require further dissection. For example, in development, Gata3 was shown to regulate HSCs by means of controlling SNS development and the secretion of HSC-supportive catecholamines [20]; however, its expression patterns in the HSC microenvironment suggest a more complex role in haematopoietic stem and progenitor cell regulation (figure 3). In the sub-aortic mesenchymal compartment, Gata3 expression is restricted to a few scattered cells on the ventral side. The fact that it

has been associated with the stem and progenitor compartment in several tissues makes it tempting to speculate that its expression marks mesenchymal stem cells. Dissecting its function in each cell type and identifying its role in HSC precursors and the HSC regulatory microenvironment is an intricate process and will require tissue-specific deletion.

In addition, the scarcity of these cells and the dynamic nature of the developing embryo add more challenges to identifying the role of Gata3 in the emerging definitive haematopoietic system. However, the availability of powerful tools such as RNA-Seq, which can now be performed on a single-cell level, will help to identify the genetic programme of these individual cell types and contribute to a better understanding of the nature and function of these cells and how Gata3 may influence their function. It is, however, very likely that the haematopoietic phenotype described in germline-deleted *Gata3*-null embryos [20,21] is a compound phenotype resulting from the effects of Gata3 deletion in the various AGM cell types.

Data accessibility. This article has no additional data.
Competing interests. We declare we have no competing interests.
Funding. Our Gata3-related work was supported by an Intermediate Fellowship from the Kay Kendall Leukaemia Fund (K.O.) and by a fellowship from the King Abdullah International Medical Research Centre (KAIMRC), Ministry of National Guard (N.Z.).

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
