## [Reviewer comments · Open Biology]

Review History

RSOB-18-0152.R0 (Original submission)

Review form: Reviewer 1

Recommendation

Accept with minor revision (please list in comments)

Are each of the following suitable for general readers?

- a) **Title**
Yes
- b) **Summary**
Yes
- c) **Introduction**
Yes

Is the length of the paper justified?

Yes

Should the paper be seen by a specialist statistical reviewer?

No

Is it clear how to make all supporting data available?

Not Applicable

Is the supplementary material necessary; and if so is it adequate and clear?

Not Applicable

Do you have any ethical concerns with this paper?

No

Comments to the Author

Although several reviews on the transcription factor GATA3 have been published to date, this manuscript by Zaidan and Ottersbach has a unique and interesting focus on the role of GATA3 in hematopoietic development, and particularly in the emergence of hematopoietic stemcells in the embryo. This review was a pleasure to read. Within the field of hematopoiesis it is comprehensive and contains an informative in-depth discussion on all the supportive evidence for our current view of the role of GATA3 (which is also nicely illustrated in Figure 3).

I only have a few minor comments.

Page 5. The sentence "Gata1 induces megakaryocyte and erythrocyte development, while preventing granulocyte-monocyte and lymphoid commitment" is quite confusing, since Gata3 is also expressed in granulocytes, including eosinophils as is clear from the next sentence. This should be rephrased. Likewise, the sentence linking Gata2 to HSC is confusing as GATA2 is e.g. also involved in mast cell development.

Page 7. When the importance of the level of GATA3 is discussed, it would be nice to include some older literature the phenotype of GATA3 overexpression, e.g. leading to thymocyte tumors (Nawijn et al., J immunol. 2001) or mast cell lineage diversion (Taghon et al., Nat Immunol. 2007).

Pages 10/11. For the role of GATA3 in breast tumors, I suggest to mention estrogen sensitivity (because it is very important in the clinic). GATA3 is a regulator of estrogen receptor expression as it binds to regulatory elements in the ERalpha locus.

Page 11. Gata3 is also involved in ear development. Although this is surely not the focus of this review, the authors may consider to mention the human condition, HDR syndrome, that is caused by GATA3 haplo-insufficiency whereby deafness is one

Review form: Reviewer 2

Recommendation

Accept with minor revision (please list in comments)

Are each of the following suitable for general readers?

a) **Title**
Yes

b) **Summary**
Yes

c) **Introduction**
Yes

Is the length of the paper justified?

Yes

Should the paper be seen by a specialist statistical reviewer?

No

Is it clear how to make all supporting data available?

Not Applicable

Is the supplementary material necessary; and if so is it adequate and clear?

Not Applicable

Do you have any ethical concerns with this paper?

No

Comments to the Author

The well-written review by Zaidan and Ottersbach focuses on the critical GATA factor GATA-3, which has not been thoroughly considered in recent reviews. Given the unique facets of GATA-3 biology and potential mechanisms, this represents a timely and important article, and only minor changes are suggested below to increase accurate of two citations.

1) "when Gata1 is expressed, Gata2 is displaced" etc – The authors cite reference 22, a review from M. Yamamoto's group. However, this finding was demonstrated by Grass et al. PNAS 2003 and developed in multiple subsequent papers in the Bresnick group (e.g., reviewed in Bresnick et al. J. Biol. Chem. 2010; Katsumura et al. Blood 2017).

2) "as Gata2 is required for their emergence". The authors cite de Pater, but not the co-published Gao et al. JEM 2013, which rigorously established a GATA-2 requirement for HSC emergence. Gao et al. also described GATA-2 regulation of Runx1 expression in the AGM, in the context of HSC emergence.

Decision letter (RSOB-18-0152.R0)

17-Oct-2018

Dear Professor Ottersbach

We are pleased to inform you that your manuscript RSOB-18-0152 entitled "The multi-faceted role of Gata3 in developmental haematopoiesis" has been accepted by the Editor for publication in Open Biology. The reviewer(s) have recommended publication, but also suggest some minor revisions to your manuscript. Therefore, we invite you to respond to the reviewer(s)' comments and revise your manuscript.

Please submit the revised version of your manuscript within 14 days. If you do not think you will be able to meet this date please let us know immediately and we can extend this deadline for you.

- 1) A text file of the manuscript (doc, txt, rtf or tex), including the references, tables (including captions) and figure captions. Please remove any tracked changes from the text before submission. PDF files are not an accepted format for the "Main Document".
- 2) A separate electronic file of each figure (tiff, EPS or print-quality PDF preferred). The format should be produced directly from original creation package, or original software format. Please note that PowerPoint files are not accepted.
- 3) Electronic supplementary material: this should be contained in a separate file from the main text and meet our ESM criteria (see <http://royalsocietypublishing.org/instructions-authors#question5>). All supplementary materials accompanying an accepted article will be treated as in their final form. They will be published alongside the paper on the journal website and posted on the online figshare repository. Files on figshare will be made available approximately one week before the accompanying article so that the supplementary material can be attributed a unique DOI.

Online supplementary material will also carry the title and description provided during submission, so please ensure these are accurate and informative. Note that the Royal Society will not edit or typeset supplementary material and it will be hosted as provided. Please ensure that the supplementary material includes the paper details (authors, title, journal name, article DOI). Your article DOI will be 10.1098/rsob.2016[last 4 digits of e.g. 10.1098/rsob.20160049].

- 4) A media summary: a short non-technical summary (up to 100 words) of the key findings/importance of your manuscript. Please try to write in simple English, avoid jargon, explain the importance of the topic, outline the main implications and describe why this topic is newsworthy.

Images

Data-Sharing

It is a condition of publication that data supporting your paper are made available. Data should be made available either in the electronic supplementary material or through an appropriate

repository. Details of how to access data should be included in your paper. Please see <http://royalsocietypublishing.org/site/authors/policy.xhtml#question6> for more details.

Data accessibility section

Sincerely,

The Open Biology Team
<mailto:openbiology@royalsociety.org>

ditage Insights by clicking on the following link: <https://www.surveymonkey.com/r/author-perspectives-on-academic-publishing-royal-society>

This should take no more than 15 minutes and you will have the opportunity to enter a prize draw. We hope these results will provide us with valuable insights we can use to improve our service.

Reviewer(s)' Comments to Author:

Referee: 1

Comments to the Author(s)

Although several reviews on the transcription factor GATA3 have been published to date, this manuscript by Zaidan and Ottersbach has a unique and interesting focus on the role of GATA3 in hematopoietic development, and particularly in the emergence of hematopoietic stemcells in the embryo. This review was a pleasure to read. Within the field of hematopoiesis it is comprehensive and contains an informative in-depth discussion on all the supportive evidence for our current view of the role of GATA3 (which is also nicely illustrated in Figure 3).

I only have a few minor comments.

Page 5. The sentence "Gata1 induces megakaryocyte and erythrocyte development, while preventing granulocyte-monocyte and lymphoid commitment" is quite confusing, since Gata3 is also expressed in granulocytes, including eosinophils as is clear from the next sentence. This should be rephrased. Likewise, the sentence linking Gata2 to HSC is confusing as GATA2 is e.g. also involved in mast cell development.

Page 7. When the importance of the level of GATA3 is discussed, it would be nice to include some older literature the phenotype of GATA3 overexpression, e.g. leading to thymocyte tumors (Nawijn et al., J Immunol. 2001) or mast cell lineage diversion (Taghon et al., Nat Immunol. 2007).

Pages 10/11. For the role of GATA3 in breast tumors, I suggest to mention estrogen sensitivity (because it is very important in the clinic). GATA3 is a regulator of estrogen receptor expression as it binds to regulatory elements in the ERalpha locus.

Page 11. Gata3 is also involved in ear development. Although this is surely not the focus of this review, the authors may consider to mention the human condition, HDR syndrome, that is caused by GATA3 haplo-insufficiency whereby deafness is one

Referee: 2

Comments to the Author(s)

The well-written review by Zaidan and Ottersbach focuses on the critical GATA factor GATA-3, which has not been thoroughly considered in recent reviews. Given the unique facets of GATA-3 biology and potential mechanisms, this represents a timely and important article, and only minor changes are suggested below to increase accuracy of two citations.

1) "when Gata1 is expressed, Gata2 is displaced" etc - The authors cite reference 22, a review from M. Yamamoto's group. However, this finding was demonstrated by Grass et al. PNAS 2003 and developed in multiple subsequent papers in the Bresnick group (e.g., reviewed in Bresnick et al. J. Biol. Chem. 2010; Katsumura et al. Blood 2017).

2) "as Gata2 is required for their emergence". The authors cite de Pater, but not the co-published Gao et al. JEM 2013, which rigorously established a GATA-2 requirement for HSC emergence. Gao et al. also described GATA-2 regulation of Runx1 expression in the AGM, in the context of HSC emergence.

Author's Response to Decision Letter for (RSOB-18-0152.R0)

See Appendix A.

Decision letter (RSOB-18-0152.R1)

29-Oct-2018

Dear Professor Ottersbach

We are pleased to inform you that your manuscript entitled "The multi-faceted role of Gata3 in developmental haematopoiesis" has been accepted by the Editor for publication in Open Biology.

Sincerely,

The Open Biology Team
mailto:openbiology@royalsociety.org

Appendix A

Response to Reviewers:

We thank the reviewers for their positive comments and for highlighting inconsistencies and missing references. We have corrected these in the manuscript, with changes shown in red. Below are also detailed responses to the individual comments made by the reviewers.

Referee 1:

Although several reviews on the transcription factor GATA3 have been published to date, this manuscript by Zaidan and Ottersbach has a unique and interesting focus on the role of GATA3 in hematopoietic development, and particularly in the emergence of hematopoietic stem cells in the embryo. This review was a pleasure to read. Within the field of hematopoiesis it is comprehensive and contains an informative in-depth discussion on all the supportive evidence for our current view of the role of GATA3 (which is also nicely illustrated in Figure 3).

I only have a few minor comments.

Page 5. The sentence “Gata1 induces megakaryocyte and erythrocyte development, while preventing granulocyte-monocyte and lymphoid commitment” is quite confusing, since Gata3 is also expressed in granulocytes, including eosinophils as is clear from the next sentence. This should be rephrased. Likewise, the sentence linking Gata2 to HSC is confusing as GATA2 is e.g. also involved in mast cell development.

We agree that these sentences were somewhat contradictory and have therefore rephrased them slightly to better reconcile these different parts. They now read as follows: “The main function of Gata1 is cell fate determination at an early branch point in the haematopoietic tree, where it induces megakaryocyte and erythrocyte development, while preventing granulocyte-monocyte and lymphoid commitment. However, it is also expressed further downstream in common lymphoid and myeloid progenitors, mast cells and eosinophils [16, 17]. The most critical role of Gata2 is the formation and maintenance of HSCs [18, 19], although it has additional functions in specific blood lineages as discussed below.”

Page 7. When the importance of the level of GATA3 is discussed, it would be nice to include some older literature the phenotype of GATA3 overexpression, e.g. leading to thymocyte tumors (Nawijn et al., J immunol. 2001) or mast cell lineage diversion (Taghon et al., Nat Immunol. 2007).

We have now included these examples of the consequences of Gata3 overexpression and have added the following sentence: “Indeed, enforced expression of Gata3 in the T cell lineage caused a maturation arrest in the cytotoxic T cell lineage and promoted the formation of thymic lymphoblastoid tumours [41], and overexpression of Gata3 at early foetal thymocyte stages redirected their differentiation towards the mast cell lineage [42].”

Pages 10/11. For the role of GATA3 in breast tumors, I suggest to mention estrogen sensitivity (because it is very important in the clinic). GATA3 is a regulator of estrogen receptor expression as it binds to regulatory elements in the ERalpha locus.

We have now also included a couple of sentences on the regulation of ERalpha expression and oestrogen-responsiveness by GATA3 in breast cancer: "The involvement of GATA3 in breast cancer, however, is complex as it was also shown to promote the growth of oestrogen-responsive tumours through direct binding to and activation of the oestrogen receptor α (ER α) gene [85]. ER α -positive tumours display a more differentiated phenotype and are generally less aggressive, which may be another reason why GATA3 expression in breast cancer is associated with a more favourable prognosis [86-88]."

Page 11. Gata3 is also involved in ear development. Although this is surely not the focus of this review, the authors may consider to mention the human condition, HDR syndrome, that is caused by GATA3 haplo-insufficiency whereby deafness is one

We have now added a section on inner ear development and HDR syndrome on page 10 to represent the full spectrum of GATA3 involvement in human disease.

Referee 2:

The well-written review by Zaidan and Ottersbach focuses on the critical GATA factor GATA-3, which has not been thoroughly considered in recent reviews. Given the unique facets of GATA-3 biology and potential mechanisms, this represents a timely and important article, and only minor changes are suggested below to increase accurate of two citations.

1) "when Gata1 is expressed, Gata2 is displaced" etc – The authors cite reference 22, a review from M. Yamamoto's group. However, this finding was demonstrated by Grass et al. PNAS 2003 and developed in multiple subsequent papers in the Bresnick group (e.g., reviewed in Bresnick et al. J. Biol. Chem. 2010; Katsumura et al. Blood 2017).

We apologise for the omission of these relevant references, which we have now added on page 5.

2) "as Gata2 is required for their emergence". The authors cite de Pater, but not the co-published Gao et al. JEM 2013, which rigorously established a GATA-2 requirement for HSC emergence. Gao et al. also described GATA-2 regulation of Runx1 expression in the AGM, in the context of HSC emergence.

Again we apologise for the omission of this important study, which we now reference on pages 5 and 6.